# Do We Store Packed Red Blood Cells under “Quasi-Diabetic” Conditions?

**DOI:** 10.3390/biom11070992

**Published:** 2021-07-05

**Authors:** Leonid Livshits, Gregory Barshtein, Dan Arbell, Alexander Gural, Carina Levin, Hélène Guizouarn

**Affiliations:** 1Red Blood Cell Research Group, Institute of Veterinary Physiology, Vetsuisse Faculty, University of Zürich, CH-8057 Zurich, Switzerland; leonidlivshts@gmail.com; 2Biochemistry Department, The Faculty of Medicine, The Hebrew University of Jerusalem, Jerusalem 91905, Israel; 3Pediatric Surgery Department, Hadassah Hebrew University Medical Center, Jerusalem 91120, Israel; arbell@hadassah.org.il; 4Department of Hematology, Hadassah Hebrew University Medical Center, Jerusalem 91120, Israel; gural@hadassah.org.il; 5Pediatric Hematology Unit, Emek Medical Center, Afula 1834111, Israel; levin_c@clalit.org.il; 6The Ruth and Bruce Rappaport Faculty of Medicine, Technion-Israel Institute of Technology, Haifa 32000, Israel; 7Institut de Biologie Valrose, Université Côte d’Azur, CNRS, Inserm, 28 Av. Valrose, 06100 Nice, France; helene.guizouarn@univ-cotedazur.fr

**Keywords:** type 2 diabetes mellitus, RBC storage, RBC transfusion

## Abstract

Red blood cell (RBC) transfusion is one of the most common therapeutic procedures in modern medicine. Although frequently lifesaving, it often has deleterious side effects. RBC quality is one of the critical factors for transfusion efficacy and safety. The role of various factors in the cells’ ability to maintain their functionality during storage is widely discussed in professional literature. Thus, the extra- and intracellular factors inducing an accelerated RBC aging need to be identified and therapeutically modified. Despite the extensively studied in vivo effect of chronic hyperglycemia on RBC hemodynamic and metabolic properties, as well as on their lifespan, only limited attention has been directed at the high sugar concentration in RBCs storage media, a possible cause of damage to red blood cells. This mini-review aims to compare the biophysical and biochemical changes observed in the red blood cells during cold storage and in patients with non-insulin-dependent diabetes mellitus (NIDDM). Given the well-described corresponding RBC alterations in NIDDM and during cold storage, we may regard the stored (especially long-stored) RBCs as “quasi-diabetic”. Keeping in mind that these RBC modifications may be crucial for the initial steps of microvascular pathogenesis, suitable preventive care for the transfused patients should be considered. We hope that our hypothesis will stimulate targeted experimental research to establish a relationship between a high sugar concentration in a storage medium and a deterioration in cells’ functional properties during storage.

## 1. Red Blood Cells Transfusion

Blood components transfusion is a common practice applied globally to more than 5% of hospitalized patients. With more than 100 million human blood units collected each year for therapeutic purposes [1], transfusion of red blood cells (RBCs) is one of the most common life-saving procedures in medicine. Depending on the severity of blood loss or anemia, packed RBC (PRBC) supplemented with a fraction of the storage medium are transfused into the recipient’s bloodstream. Blood donations for transfusion are routinely stored as PRBC, for up to 35 or 42 days, depending on the storage medium [2,3]. However, in recent years, there has been increasing concern about the safety and efficacy of PRBC transfusion [4,5]. A growing number of studies show that it can cause damage rather than benefit its recipients [5,6,7,8].

Despite numerous attempts (such as improved donor screening and patient blood management) to minimize post-procedural risks [8,9], there still exists a long list of possible transfusion-associated immediate or late-onset deleterious effects on the recipient’s health. Among others, transfusion-related febrile non-hemolytic reaction, mild to moderate allergic reaction, and delayed hemolytic transfusion reaction are the most commonly reported [10]. In addition, volume overload, adverse immunomodulation [11], and multiple types of endothelial dysfunction [12,13,14] have also been discussed in the literature. Moreover, patients who receive PRBC transfusions have longer durations of hospital stay and higher rates of infection, spend more time in the intensive care, and are at a higher risk for acute respiratory distress syndrome (ARDS) [15,16,17]. Thus, Hopewell et al. [17], in their systematic review, analyze observational studies published from 2006 to 2010 and conclude that there is a consistent effect of blood transfusion on recipients’ mortality, and that the PRBC associated risks are dose-dependent [18,19].

Several studies have linked the occurrence of complications after PRBC transfusion to the RBC storage lesion [20,21,22,23,24]. However, according to current blood bank regulations, only the immunological characteristics of the stored units (i.e., antibody screening, blood typing, and immune compatibility with the recipient plasma), and the presence of blood-borne infectious agents are consistently tested, while cell functionality parameters are not routinely controlled. In everyday blood banking practice, the storage duration is considered as the primary criterion for inventory management, and except for special cases, such as newborns or multi-transfused patients, the PRBC units are supplied primarily according to the first-in-first-out (FIFO) principle. In this case, the “calendar age” of the unit is accepted as a master parameter, while the actual quality of the cells provided for transfusion, specifically their capacity to provide the expected transfusion outcome, is not addressed. In their review, Koch et al. [25] examined “the evidence of the so-called storage lesion for RBC” and suggested that the storage duration (calendar age) is not an appropriate measure of the PRBC quality, and that “a functional measure of stored RBC quality—real age—may be better than calendar age”. They suggested that using “metrics to measure temporal changes in the quality of the stored RBC product may be more appropriate than the 42-day expiration date” and proposed several potential markers of the RBCs “real age”.

## 2. Red Blood Cells Aging (In Vitro and In Vivo)

The aging of red blood cells (RBCs) in vivo and in vitro represents a key biomedical issue. Human RBCs have an approximate lifespan of 120 days in vivo [26,27], which is much longer than the 35–42-day shelf life of RBC concentrates stored under blood bank conditions. Human RBCs survive to about the same age, which implies the existence of a molecular countdown that triggers a series of changes leading to their removal by the reticuloendothelial system. It is important to emphasize that although there is much in common in the process of RBC aging under both conditions, a significant difference in the in vivo and in vitro aging of erythrocytes has been documented.

### 2.1. In Vivo Aging

Several senescence markers are known that tag RBCs as senescent and prime them for clearance from the blood stream [27]. It is currently believed that metabolic changes involving ATP depletion and progressive oxidation, along with a reduction in mechanical stability and increase in cytosolic density and rigidity, are the primary processes triggering clearance [28,29]. These changes are mainly mediated by a progressive decrease in the activity of glycolytic and anti-oxidative enzymes, and by the loss of membrane surface resulting from shear stress. Proteolysis and oxidation are caused by Ca^2+^ uptake bouts, triggered by shear stress and exposure to inflammatory factors and myokines. Oxidation promotes the changes in tyrosine phosphorylation and clustering of band-3 protein [30], which is subsequently recognized by the naturally occurring antibodies [30,31]. Loss of membrane and an increase in cell density and cytosol’ viscosity due to an elevation in the mean corpuscular hemoglobin concentration (MCHC) are observed with aging [32,33,34]. Finally, the inability to maintain Na^+^/K^+^ and Ca^2+^ gradients contributes to cell dehydration and increases the proteolytic activity of Ca^2+^-dependent protease calpain [35,36,37,38]. Loss of cell water and an increased MCHC make the cells more rigid and susceptible to mechanic clearance within the splenic sinuses. Furthermore, aging-related vesiculation of RBC membrane [39] induces a decrease in the erythrocyte surface-to-volume ratio [40] and, following transformation from discoid to spherocyte/echinocyte shape [41,42], an accelerated spleen clearance of cells [43,44]. A satisfactory understanding of which are the master regulators of RBC aging, and which are the secondary events, and of the complex interaction between these processes, is still lacking.

In this view, special attention will be focused on the plasma glycemic state’s effect on RBC aging processes. Because of glucose transport specificity via the insulin-independent GLUT1 [45,46], glucose concentration inside the RBCs is directly related to its content in the exterior solution (the blood plasma or storage solution). Therefore, a prolonged exposure to external hyperglycemic conditions may result in undesired modification of RBC constituents (such as non-enzymatic irreversible glycation of RBC proteins and oxidative stress) [47,48] with a consequent negative effect on RBC functionality and life span. Despite some evidence to the contrary [49,50], most studies directly associate hyperglycemia with a diminished RBC lifespan [51,52,53,54,55]. Several hypotheses have been proposed to clarify this phenomenon; among others, an abnormal activity of glycated RBC membrane proteins, resulting in a reduced negative surface electrical charge, an increased fluidity, an abnormal lipid peroxidation, and a high adherence of RBCs to endothelial cells have been offered to explain how abnormal hyperglycemia may accelerate RBC aging [56,57,58,59,60]. This topic has been discussed in detail in the literature and analyzed in several reviews [46,61,62].

### 2.2. In Vitro Aging (Cold-Storage)

Before considering the problem of the in vitro aging of cells during their storage in a blood bank, it is necessary to take into account that all cells in the donated unit already have a certain (donor specific) “history of aging” in vivo. Donated blood contains RBCs that range from 0 to 120 days of age; “young” cells are relatively stable under cold-storage conditions, but the “old” ones are more sensitive to storage-related stress. It is well documented that the damage caused to the “old” fraction of RBC unit during its cold storage is much more intense than to the “young” fraction [42,63,64,65,66].

During cold storage, in addition to their aging in vivo, RBCs undergo slow detrimental changes, collectively termed “storage lesion”. Oxidative stress and ATP-depleted metabolism are the main driving forces in the development of this lesion. Storage-related processes lead to significant metabolic and structural changes in the erythrocytes, including global biochemical and biophysical alterations, remodeling of cell membrane structure, and of cytoplasm composition (see also Table 1).

The best-known changes include ATP and DPG depletion [32]; loss of cellular antioxidant capability [296,297]; changes in the K^+^ and Na^+^ concentrations [298,299]); Ca^2+^ influx [282]; loss of membrane and skeleton proteins [300,301]; loss of membrane lipids and changes in their in/out distribution; massive vesicle generation [302,303]); oxidation and remodeling of skeleton proteins [304]; clustering of the band-3 proteins [305]; and more. Visually, these changes are expressed as an alteration of the cell shape [287], i.e., transformation from a discoid form to echinocyte.

Some of these changes are interrelated and initiate a cascade of biochemical and structural alterations, which lead to impairment in the RBC functionality, specifically an alteration in the biophysical/mechanical properties of the cells. Therefore, in order for the RBC to become rigid [275,306,307,308,309] and fragile [301,310,311], having lost its ability to deform and survive in the bloodstream after transfusion [312,313,314], multiple changes listed above need to occur.

## 3. Sugar as a Potential Factor of RBC Lesion

The literature discusses the possible causes and mechanisms leading to the above changes. However, paradoxically, one of the possible factors leading to PRBCs lesions is the storage medium’s high glucose content, required to provide energy to maintain red cell viability and functionality throughout the entire weeks’ long storage. The concentration of sugar (dextrose or glucose) in different storage media (see Table 2) varied greatly [82,315,316,317,318,319]. These concentrations are significantly higher than the physiological ones (around 5 mM). Similar to in vivo chronic hyperglycemic conditions, the prolonged exposure to such external hyperglycemic conditions during the cold storage may result in significant abnormalities in RBC hemodynamic and metabolic properties [73]. HbA1c, the glycosylated form of the most abundant RBC protein, adult hemoglobin, is used to reflect the average blood glucose level over the preceding 60 days [320] and thus may serve as a measure of the glycemic status of the stored RBCs. The scarce and inconsistent data describing the change of HbA1c concentrations within the packed RBC during prolonged storage do not provide an unequivocal answer whether the effect of glycosylation is harmful to the stored RBCs [98,99,101,103,321,322,323]. This lack of data has prompted the conclusion that HbA1c values of patients receiving transfusions must be considered uninterpretable [324]. In parallel, the excess of glucose in the storage medium may lead to an elevated metabolism by RBC accompanied by increased lactate release and the medium’s acidity. Consequently, high acidity may result in partial hemolysis associated with enhanced transmembrane ion and water transport. Most of these processes have been previously reported [73].

This mini-review compares the biophysical and biochemical changes in the RBCs that occur under cold-storage conditions and in patients with non-insulin-dependent diabetes. Diabetes mellitus is a group of complex and multifactorial metabolic diseases affecting almost half of a billion individuals in the world (IDF Diabetes Atlas. Ninth Edition; 2019.). Insulin resistance associated with blood hyperglycemia and hyperlipidemia [325,326] induces a decreased glucose utilization by most tissues and impaired insulin secretion by the pancreas and hepatic glucose production. Hyperglycemia is probably a key trigger for the disease pathogenesis and the progression of diabetic complications. In addition to the acute stress for patients, hyperglycemia is linked to the development of long-term diabetic complications, which include nephropathy, retinopathy, neuropathy, cardiovascular disease, peripheral vascular problems, tooth and gum disease, and sexual dysfunction [327,328,329,330,331]. This is the main reason that preventing end-organ damage by preventing hyperglycemia (i.e., keeping blood glucose levels near the normal range) is the most crucial part of diabetes treatment.

Organ, and specifically end-organ, damage can be provoked by the disturbance of blood circulation/microcirculation [332,333,334] and, respectively, by RBC abnormalities [335,336]. The role of RBCs morphology and deformability in blood microcirculation has been strongly documented [277,286,337,338]. Unsurprisingly, most macro and, mainly, microvascular abnormalities in diabetic patients are directly associated with RBCs’ pathological features, a primary and most abundant target for glucose exposure [119,284,339,340,341]. Table 1 summarizes several (but, necessarily, not all) of the RBC features altered with the progression of non-insulin-dependent diabetes (NIDDM), which may associate with the aforementioned complications.

During cold storage, PRBCs undergo changes induced by (1) high glucose-mediated exposure, resultant glycation, and associated ROS elevation [104]; (2) by products of RBC metabolism, including elevated external acidity [70,73,75,104,157,200,342,343,344]; and (3) free hemoglobin levels elevation due to the partial hemolysis of stored red cells. Given the well-described corresponding RBC alterations in NIDDM and during cold storage, we may regard the stored (especially long-stored) RBCs as “quasi-diabetic.” Keeping in mind that these RBC modifications may be crucial for the initial steps of microvascular pathogenesis [345], suitable preventive care for the transfused patients should be considered.

The patients requiring multiple and frequent transfusions as a therapeutic intervention are especially at risk from such RBCs modifications. This relates to several pathologies associated with severe anemia, such as some blood cancers and chronic kidney disease, and most congenital hemoglobinopathies.

For many years, chronic kidney disease (CKD) has been considered as one of the main public health problems. For example, only in the USA, ~15% of adults suffer various forms of CKD, and, amongst them, around 800,000 people (i.e., 2 in every 1000) are currently living with end-stage renal disease (ESRD) (https://www.cdc.gov/kidneydisease/pdf/Chronic-Kidney-Disease-in-the-US-2021-h.pdf). In China, with its population of almost 1.5 billion, the prevalence of chronic kidney disease in the population aged 18 years or older is less, but still very significant (10.8%) [346]. One of the common complications of moderate-to-severe CKD is anemia [347,348,349,350,351] and its correction is known to have beneficial effects on cardiac function [352,353,354]. For decades, RBC transfusion has been considered a main therapeutic solution for these patients, especially before the development of erythropoiesis-stimulating agents [355,356,357,358,359]. Numerous studies dealt with possible risks and complications of chronic transfusion in CKD patients, such as the transmission of blood-borne diseases, iron and potassium overload [360,361,362], and also sensitization (the development of antibodies to foreign antigens) [363,364]. To the best of our knowledge, no study has examined the interrelated connections between the transfusion frequency, glucose-induced changes of blood cells, and further complications in the CKD patients. This is in view of the peripheral neuropathy [365,366,367,368,369,370,371], retinopathy [372,373,374,375,376,377,378,379,380,381], as well as other macro-and microvascular complications associated with hyperglycemia that are well known in patients with CKD. It is currently impossible to evaluate the part of the storage-associated alteration of PRBC on the development of CKD complications, especially the role of high glucose in these processes.

The standard of care for patients with severe hemoglobinopathies (including Sickle cell anemia and major β-thalassemia) is mainly based on PRBC transfusions [382,383]. The frequency of transfusions may reach one per every two weeks and even more often. In other words, a large fraction of “quasi-diabetic” RBC with the hemodynamic and metabolic alterations are frequently infused into the bloodstream of these patients. Despite the fact that immediate as well as accumulative deleterious effects of blood transfusion are hard to evaluate, the development of microvascular complications such as peripheral neuropathy [384,385,386,387,388,389,390,391,392,393,394,395,396,397], retinopathy [398,399,400,401,402,403,404,405,406,407,408,409,410], leg ulcers [411,412,413,414,415,416,417,418,419,420,421,422], and kidney dysfunction [34,423,424,425,426,427,428,429,430,431,432,433,434,435,436,437,438,439,440,441,442,443,444] are well known in these patients. It is accepted in the literature that these changes may occur mainly due to iron overload secondary to blood transfusion or as a side-effect of the treatment with iron chelators. Barshtein et al. [312,445] demonstrated that transfusion of PRBC with a high portion of low-deformable cells causes decreased skin-blood-flow in beta-thalassemia patients [312,445]. Nevertheless, as a possible support for our hypothesis, several authors recently reported an increase in plasma levels of advanced glycation end-products (AGE) and their potential contribution to the pathophysiology of chronic hemolysis and organ damage in major β-thalassemic and sickle cell patients [446,447,448,449].

## 4. Limitation

The authors want to draw readers’ attention to the fact that the approach they have presented has a significant limitation. As is well known, the properties of cells are determined by a wide range of factors. These include the mineral and protein composition of the medium, temperature, and external mechanical influences. There is a significant difference between the existence of erythrocytes in the PRBC unit and circulating blood. For example, under storage conditions, cells are suspended in storage-medium rather than plasma. In addition, under storage conditions, the cells are immobile at a temperature of 4 °C, while in the body, they are constantly circulating, and the blood temperature is 37 °C. In the presented text, from all the influencing factors, we singled out for consideration one single indicator, a high concentration of sugar. This undoubtedly dilutes the proposed conclusions. However, our analogies (in the two discussed states) of the behavior of erythrocytes indicate the adequacy (qualitative) of the presented approach.

## 5. Conclusions

In conclusion, we speculate that much of the damage to packed red blood cells during cold storage can be triggered by the presence of glucose or dextrose in the storage-medium, with these components acting in this respect as “double agents”. As far as we know, the fact that the storage medium is a concentrated sugar solution has not yet been considered a factor capable of affecting PRBCs properties negatively. We hope that our hypothesis will stimulate targeted experimental research to establish a relationship between a high concentration of sugar in a storage medium and a deterioration in cells’ functional properties during storage. Moreover, the almost uninvestigated alterations in glucose uptake, glycolysis, and associated regulative processes during cold storage should certainly be addressed in the future. In our opinion, these data are necessary to optimize storage medium formulation to reduce damage to red blood cells occurring during prolonged storage.

## Figures and Tables

**Table 1 biomolecules-11-00992-t001:** Comparison between some features of RBC from non-insulin-dependent diabetic patients and stored units.

	T2 Diabetes	RBC Storage
Elevated Hemolysis/Free Heme	[67]	[68,69,70,71,72,73,74,75,76,77,78,79,80,81,82,83,84,85]
Elevated Membrane Phosphatidylserine exposure	[86,87]	[88,89,90,91,92]
Elevated HbA1C	[93,94,95,96,97]	[73,98,99,100,101,102,103]
Elevated Intra-RBC ROS concentration	[100,102]	[71,104,105]
Decreased levels/activity of RBC GSH and other antioxidant systems	[102,106,107,108,109,110,111,112,113,114,115,116,117,118,119,120,121,122,123,124,125,126,127,128,129,130,131,132,133,134]	[69,104,135,136,137,138]
Elevated Intracellular AGE	[58,102,139,140,141,142,143,144,145,146,147]	[148,149]
Abnormalities in Nitric oxide signaling and decreased RBC nitric oxide synthase (RBC-NOS) activity	[150]	[84,151]
Decreased 2,3 DPG level	[152,153,154,155]	[33,70,73,88,156,157,158,159,160,161,162,163]
Abnormalities in Na/K levels and decreased Na+/K+-ATPase activity	[128,141,164,165,166,167,168,169,170,171,172,173,174,175,176]	[70,72,177,178,179,180,181,182,183,184]
Ca2+ intracellular accumulation and/or decreased Ca2+ ATPase activity	[141,172,185,186,187,188,189,190,191]	[73,75,192,193]
Decreased intracellular ATP level	[154,172]	[68,72,73,75,77,80,81,82,83,88,156,157,162,194,195,196,197,198,199,200,201,202,203,204]
Elevated intra-RBC protein oxidation	[102,118,145,205,206,207,208,209,210,211,212]	[34,71,73,79,135,137,162,179,181,213,214,215,216,217,218,219,220,221,222,223,224,225,226,227,228,229,230,231,232,233]
Elevated lipid peroxidation	[106,114,115,126,127,141,145,210,234,235,236,237,238,239]	[79,135,162,179,215,221,240,241]
Elevated Poly Unsaturated Fatty Acid (PUFA) oxidation	[127,242]	[73,148,243]
Decreased ATP release from RBC	[244,245,246,247]	[199,248,249]
Decreased intra-RBC NADPH	[110,111,247]	[104]
Decreased RBC deformability	[59,187,211,250,251,252,253,254,255,256,257,258,259,260,261,262,263,264,265,266,267,268]	[70,73,156,197,216,269,270,271,272,273,274,275,276,277]
Elevated RBC adhesion	[278,279,280]	[275,281,282]
Elevated RBC aggregability	[262,283,284,285,286]	[162,275,287]
Elevated release of Extracellular vesicles (EVs)	[288]	[69,72,73,156,192,289,290,291,292,293,294,295]

**Table 2 biomolecules-11-00992-t002:** Sugar concentration in different storage media [75,309,310,311,312,313].

Storage Medium	Sugar Concentration, mM	
	Glucose	Dextrose
MAP	40	-
CPDA-1	177	-
AS-1	111	
AS-3	-	55
AS-7	80	-
SAGM	45	-
PAGGSM	47	

## Data Availability

Not applicable.

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
