# Peer review of "Do We Store Packed Red Blood Cells under “Quasi-Diabetic” Conditions?"

_biomolecules, 2021, doi:10.3390/biom11070992_

Round 1
Reviewer 1 Report
The opinion manuscript by Livshits et al brings the reader’s attention to the fact that RBCs stored in high glucose solutions, currently used for blood banking, and RBCs from diabetic patients, who bathe in high glucose plasma during their lifespan, have similar altered properties. They explain that the known problems associated with diabetic RBCs in diabetic patients have not been considered with stored RBCs although they are stored in diabetic-like conditions for up to 42 days. They also underline that transfusion of diabetic-like RBCs may cause negative secondary effect in populations with regular transfusions such as CKD patients and patients with congenital hemoglobinopathies. They finally propose that experimental approaches should be designed to explore these potential problems. The manuscript is well written and provides an original and very well documented point of view. I provide comments and questions to help the authors clarify some sections.
General comments
- Are there alternatives to storing RBC in high glucose solutions? If high glucose do indeed accelerate RBC storage lesion, what do we do?
- Add subtitles to help readers follow reasoning?
Specific comments
- Page 1, Abstract: The focus of the abstract is mostly on introducing the reader to potential deleterious effects of transfusing stored RBCs and could describe more precisely the original content of the manuscript.
- Page 3, line 94-96: Could not find a demonstration of calpain activation by Ca2+ in reference 27.
- Page 3, lines 109-110: Authors write: “Despite some evidence of the contrary, most studies directly associate hyperglycemia with a diminished RBC lifespan. The reduced 24h transfusion recovery of long-stored RBC is well documented (suggesting that only a subpopulation of RBC is sufficiently altered during storage to be removed from circulation in the hours following transfusion). Are there any studies reporting a reduced lifespan of RBC stored in diabetic-like conditions? Would the whole RBC population be affected by storage in diabetic-like conditions or only a subpopulation?
- Page 3, lines 123-124: Authors write “It is well documented that the damage caused to the old fraction of RBC unit during its cold storage is much more intense than to the young fraction”, but they only provide one recent reference in English (38). Consider adding other references to support this claim.
- Page 3 and 4, Table 1: Are there any studies reporting altered RBC morphology in diabetic patients?
- Page 6, lines 181-183: Authors write “Unsurprisingly, most macro- and, mainly microvascular abnormalities in diabetic patients are directly associated with RBC’s pathological features…” This sentence contains a key argument of the manuscript: that some vascular abnormalities are associated with RBC altered properties. Consider developing this argument to better support the proposition that RBC alterations are causing some of the diabetic complications.
- Page 6, lines 219-220: Please consider revising this sentence: “The rate of transfusions may…”
- Page 7, lines 235-241: Please consider clarifying the arguments presented in this paragraph.
Author Response
Livshits et al.
Do we store packed red blood cells under "quasi-diabetic" conditions?
Response to Editor:
Thank you for your interest in our work and the pertinent and helpful comments to improve the paper. We have carefully considered the reviewer's points and have attached a point-by-point response to all concerns, which are now also addressed with new text in the revised manuscript.
Response to Reviewer#1:
General comments
- Are there alternatives to storing RBC in high glucose solutions? If high glucose does indeed accelerate RBC storage lesion, what do we do?
We thank the reviewer for these questions.
The possible and most logical solution is to find the most optimal but minimal sugar dose that will be optimal for the RBC storage.
As another possible solution, the supplementation of the agents with specific anti-glycation activity into the storage solution may be suggested. Specifically, persistent hyperglycemia and oxidative stress accelerate the formation of AGEs 1,2. In this process, both long- and short-lived proteins become affected by advanced glycation; most importantly, their structure and function are adversely altered. Anti-glycation therapies are considered a preventive strategy against the formation of AGEs, generation of free radicals through autoxidation of glucose and glycated proteins, oxidative stress, and its consequences in diabetic complications 3. However, in view of the specific scope of this review, where we suggest possible common biological mechanisms of RBC lesion in the storage and in diabetes only, we have decided to abandon possible assumptions and speculations regarding the therapy.
2. Add subtitles to help readers follow reasoning?
We thank the reviewer for the recommendation and added relevant subtitles to the text.
Specific comments
3. Page 1, Abstract: The focus of the abstract is mostly on introducing the reader to potentially deleterious effects of transfusing stored RBCs and could describe more precisely the original content of the manuscript.
We thank the reviewer for the comment and change the Abstract content in accordance.
4. Page 3, lines 94-96: Could not find a demonstration of calpain activation by Ca2+ in reference 27.
We are thankful for this comment. We correct our mistake by discharging the originally wrong reference. As well, several other references supporting the statement are cited (see in the text):
5. Page 3, lines 109-110: Authors write: "Despite some evidence of the contrary, most studies directly associate hyperglycemia with a diminished RBC lifespan. The reduced 24h transfusion recovery of long-stored RBC is well documented (suggesting that only a subpopulation of RBC is sufficiently altered during storage to be removed from circulation in the hours following transfusion). Are there any studies reporting a reduced lifespan of RBC stored in diabetic-like conditions? Would the whole RBC population be affected by storage in diabetic-like states or only a subpopulation?
In the cited references [47-51], the authors pointed out, amongst others, the decreased lifespan of RBCs in diabetic subjects. However, in Biziak et al (Ref. 48), Brinkmann (Ref.49), Mazzanti (Ref.52), Kamada et al (Ref.55) and few others, the specific RBC subpopulations (i.e., young and old RBCs, separated on Percoll density gradient) were examined. They showed more severe rheological and biochemical alterations in older RBC compared to younger RBC. Therefore, we may speculate that the storage in diabetic-like conditions may primarily affect the older RBC subpopulation, as we mentioned below. However, this issue requires more deep investigation in the future.
6. Page 3, lines 123-124: Authors write “It is well documented that the damage caused to the old fraction of RBC unit during its cold storage is much more intense than to the young fraction”, but they only provide one recent reference in English (38). Consider adding other references to support this claim.
We add a few, most current references (Ref 64 - 66).
7. Page 3 and 4, Table 1: Are there any studies reporting altered RBC morphology in diabetic patients?
Several articles are reporting altered RBC morphology in diabetic patients, for example, Buys et al (Ref. 253 at our Review).
8. Page 6, lines 181-183: Authors write “Unsurprisingly, most macro- and, mainly microvascular abnormalities in diabetic patients are directly associated with RBC’s pathological features…” This sentence contains a key argument of the manuscript: that some vascular abnormalities are associated with RBC altered properties. Consider developing this argument to better support the proposition that RBC alterations are causing some of the diabetic complications.
We are thankful for this comment. We added relevant remarks to the revised text. (Lines 191 - 194)
9. Page 6, lines 219-220: Please consider revising this sentence: “The rate of transfusions may…”
We are thankful for this comment. We correct our mistake in the revised text. (Line 234)
10. Page 7, lines 235-241: Please consider clarifying the arguments presented in this paragraph.
We are thankful for this comment. We have rewritten the corresponding paragraph. in to the revised text. (Lines 191 - 194).
Bibliography
- Forbes JM, Cooper ME. Mechanisms of diabetic complications. Physiol Rev. 2013;93(1):137-188.
- Fu MX, Wells-Knecht KJ, Blackledge JA, Lyons TJ, Thorpe SR, Baynes JW. Glycation, glycoxidation, and cross-linking of collagen by glucose. Kinetics, mechanisms, and inhibition of late stages of the Maillard reaction. Diabetes. 1994;43(5):676-683.
- Peppa M, Vlassara H. Advanced glycation end products and diabetic complications: a general overview. Hormones (Athens). 2005;4(1):28-37.

Reviewer 2 Report
The review indicates important issues about storage of the RBCs for blood transfusion under conditions with high concentration of sugar in a storage medium and possible effect on the RBC properties. I recommend the article for publication with only several corrections: 1. page 2 line 92: the brackets ")" must be removed 2. I recommend to modify the table 1 title as "Table 1. Comparison between some features of RBC from non-insulin-dependent diabetic patients and stored unitsAuthor Response
Livshits et al.
Do we store packed red blood cells under "quasi-diabetic" conditions?
Response to Editor:
Thank you for your interest in our work and the pertinent and helpful comments to improve the paper. We have carefully considered the reviewer's points and have attached a point-by-point response to all concerns, which are now also addressed with new text in the revised manuscript.
Response to Reviewer#2:
The review indicates important issues about storage of the RBCs for blood transfusion under conditions with high concentration of sugar in a storage medium and possible effect on the RBC properties.
I recommend the article for publication with only several corrections:
1. page 2 line 92: the brackets ")" must be removed.
2. I recommend to modify the table 1 title as "Table 1. Comparison between some features of RBC from non-insulin-dependent diabetic patients and stored units
We thank the reviewer for a helpful comment and change the revised version of the manuscript following his recommendation.
